# Development and Use of a Kinetical and Real-Time Monitoring System to Analyze the Replication of Hepatitis C Virus

**DOI:** 10.3390/ijms23158711

**Published:** 2022-08-05

**Authors:** Xiaoyu Li, Masahiko Ito, Haruyo Aoyagi, Asako Murayama, Hideki Aizaki, Masayoshi Fukasawa, Takanobu Kato, Takaji Wakita, Tetsuro Suzuki

**Affiliations:** 1Department of Microbiology and Immunology, Hamamatsu University School of Medicine, Shizuoka 431-3192, Japan; 2Department of Virology II, National Institute of Infectious Diseases, Tokyo 162-8640, Japan; 3Department of Biochemistry and Cell Biology, National Institute of Infectious Diseases, Tokyo 162-8640, Japan

**Keywords:** hepatitis C virus, genome replication, bioluminescence profile, real-time monitoring, kinetical analysis

## Abstract

In microbiological research, it is important to understand the time course of each step in a pathogen’s lifecycle and changes in the host cell environment induced by infection. This study is the first to develop a real-time monitoring system that kinetically detects luminescence reporter activity over time without sampling cells or culture supernatants for analyzing the virus replication. Subgenomic replicon experiments with hepatitis C virus (HCV) showed that transient translation and genome replication can be detected separately, with the first peak of translation observed at 3–4 h and replication beginning around 20 h after viral RNA introduction into cells. From the bioluminescence data set measured every 30 min (48 measurements per day), the initial rates of translation and replication were calculated, and their capacity levels were expressed as the sums of the measured signals in each process, which correspond to the areas on the kinetics graphs. The comparison of various HuH-7-derived cell lines showed that the bioluminescence profile differs among cell lines, suggesting that both translation and replication capacities potentially influence differences in HCV susceptibility. The effects of RNA mutations within the 5′ UTR of the replicon on viral translation and replication were further analyzed in the system developed, confirming that mutations to the miR-122 binding sites primarily reduce replication activity rather than translation. The newly developed real-time monitoring system should be applied to the studies of various viruses and contribute to the analysis of transitions and progression of each process of their life cycle.

## 1. Introduction

In microbiological research, it is essential to understand the time course of each step in the lifecycle of a pathogen and how the host cell environment changes over time after infection. However, analyses of temporal changes in the lifecycles of viruses and other pathogens have been performed based on endpoint assays in which materials, including cell culture supernatants and lysates, are collected over time and analyzed for viral gene products or reporter activities. In this study, we analyzed that the virus lifecycle using a real-time recording system of reporter activity using a photomultiplier, which is a highly sensitive luminescence detector, capable of continuously detecting weak luminescence over time in culture settings without sampling. In the real-time monitoring of viral replication, the activity of luciferase, which has a relatively short half-life, is used as an indicator, and the amount of bioluminescence is measured in the culture system in which the substrate luciferin is added. This system does not require the preparation of numerous samples to measure each time point to observe alterations in kinetic bioluminescence, providing more accurate analyses of changes in cells over time compared to conventional methods.

Here, we used this system for time-course analysis of RNA replication of hepatitis C virus (HCV) and found that the system could measure the process of viral transient translation to RNA replication continuously after the introduction of synthesized viral transcripts, leading to a new mathematical model for subgenomic HCV RNA replication. HCV is an enveloped virus with a single-stranded RNA genome of positive polarity that belongs to the *Hepacivirus* genus within the *Flaviviridae* family. Chronic HCV infection is one of the leading causes of chronic liver disease, including cirrhosis and hepatocellular carcinoma [1,2]. Experimental models based on reverse genetics enable us to study specific aspects of the HCV lifecycle, such as viral RNA replication with a subgenomic replicon and the early phase from entry into cells through viral replication using HCV single-round infectious particles [3,4]. The subgenomic replicon is based on a bicistronic RNA construct as its typical design, which drives a reporter gene, such as luciferase from the HCV internal ribosomal entry site (IRES), while the HCV non-structural proteins (NS3-NS5B) are expressed from an encephalomyocarditis virus (EMCV) IRES. The transfection of in vitro-transcribed RNA into human hepatoma HuH-7-derived cells results in transiently self-replicating HCV RNA in the cells, and quantitative analysis of the reporter expression can be measured to monitor the virus replication.

## 2. Results

### Development of a Real-Time, Kinetic Monitoring System of HCV Replicon Activity

The replication of the HCV subgenomic replicon RNA based on the JFH-1 isolate carrying a synthesized green-emitting luciferase (SLG) gene (SGR-JFH1/WT/SLG; WT replicon) (Figure 1A) was monitored every 30 min (48 measurements per day) by measuring bioluminescence in the presence of luciferin in a HuH-7 subclone, HuH-7T1, that supports efficient HCV production (Figure 1B, black line). A sharp peak in the luminescence signal was detectable at around 3 h post-RNA electroporation (hpe), followed by a rising large, broad peak initiated at 20 hpe. The luminescence started to decrease at the middle of the cell growth phase (Figure 1B, green line). Based on independent measurements of three culture samples under the same conditions, the mean and coefficient of variation (CV) values were summarized, indicating that the CV was less than 0.18 throughout the culture period (Appendix A). Time-course changes in the large and broad peak of the signal were analogous to endpoint analyses of the expression of HCV NS5A, which is considered to a better indicator of viral replication than the amount of intracellular HCV RNA because it can be distinguished from the viral RNA that is not replicating after transduction (Figure 1C). In contrast, cells expressing a replication-defective replicon with substitution mutations in the active center of NS5B RNA-dependent RNA polymerase (RdRp) (SGR-JFH1/GND/SLG; GND replicon) displayed only a signal peak around 3 hpe, which coincides with the first peak for the WT replicon, but no further luminescence at later time points (Figure 1B, blue line). These findings suggest that the small peak at an early time point and another large peak observed in cells expressing the WT replicon, respectively, correspond to viral transient translation and replication. Although treatment with La peptide (LAP), which is an HCV translational inhibitor that binds to the IRES in the 5′ untranslated region (UTR) of the viral genome [5], resulted in marked inhibition of the luminescence signal, including the 1^st^ signal peak, this inhibition was not observed in cells treated with Y23Q LAP, a mutant with defective binding (Figure 2A). Only the large luminescence peak, which is assumed to indicate replication, but not the 1st peak was inhibited by the NS5A-specific inhibitor daclatasvir (DCV) (Figure 2B). HCV is known to utilize host membranes to generate the replication organelles, mainly consisting of double membrane vesicles (DMVs), which are a central hub for the synthesis of viral RNA. To determine how the number of cytoplasmic DMVs fluctuate as the WT replicon replicates, HuH-7T1 cells were collected over time after RNA electroporation and the number of cells containing DMVs was measured by transmission electron microscope (TEM) analysis (Figure 3A,B). The comparison of the results with the HCV replication profile (Figure 1B) showed that changes in the number of DMV-containing cells were observed from 20 hpe, consistent with an increase in replication levels in the early stages of HCV replication, such as 20–60 hpe. In contrast, during the late replication phase around 160–190 hpe, the number of cells containing DMV decreased to less than a quarter of the peak (at 57 hpe), despite the maintenance of high levels of the viral replication.

In this developed system, we confirmed that the bioluminescence pattern corresponding to HCV RNA replication was similar when D-luciferin, a substrate for luciferase, was added once at the beginning of the cell culture and when luciferin was added daily to the culture (Appendix A), which can be interpreted as an excess amount of luciferin in the culture medium throughout the culture period. It is known that the efficiency of luciferin uptake into cells is mainly regulated by the expression of the organic anion transporter 1 (OAT1) molecule [6]. Considering the possibility that cell proliferation affects OAT1 gene expression, we measured OAT1 mRNA in HuH7T1 cells in proliferative (24 and 48 h of cultures) and nearly confluent (72 h culture) states and found that OAT1 expression levels were comparable in all cell phases (Figure 1D). It appears that, at least in HuH7-derived cells used for HCV replication experiments, differences in culture conditions, such as cell proliferation phases, have no or little effect on the efficiency of luciferin incorporation into cells.

Next, the luminescence patterns for the viral translation and replication were compared among (1) HuH-7-derived subcloned cell lines and (2) replicons derived from various HCV genotypes (GTs). Based on kinetical measurements of luminescence, the initial rates for the transient translation and replication can be taken as representing the rate of signal change during their initial or early phase. To quantify the efficiency of HCV translation and replication, the initial rate of transient translation (Ri(T)), the sum of luminescence signals during transient translation (Ss(T)), the initial rate of replication (Ri(R)) and the sum of luminescence signals during replication (Ss(R)) (Appendix A) were calculated. As shown in Figure 4A and Table 1, Ri(T) and Ss(T) values in Huh7.5.1-8 and HuH-7-Scr were larger than those in HuH-7T1 and HuH-7-JCRB. In contrast, the time course change of replication and Ri(R) value were variable among the cell lines, and the Ss(R) value in HuH-7T1 was the highest, followed by Huh7.5.1-8, HuH-7-Scr, and HuH-7-JCRB. Accordingly, the highest luminescence signal of the replication peak was detected in Huh7.5.1-8 at the earliest time point (34 hpe). These findings are largely consistent with those of previous studies [7,8]. Judging from the flow cytometric analysis, which measured the percentage of GFP-positive cells after GFP RNA electroporation, the four cell lines used in the experiment showed no difference in the efficiency of gene transfer into cells by HCV RNA electroporation (Figure 4B and Appendix A). Cell viability assay showed that none of the cells were damaged by RNA electroporation to any significant degree (Figure 4C).

When comparing the translation and replication activities among replicons from GT-1b (Con1) [9], -2a (JFH-1) [10], and -3a (S310) [11] in Huh7.5.1-8 cells, the luminescence signal of the replication peak generated from the GT-2a replicon was markedly higher compared to the others, ~30- and ~700-fold higher than GT-1b and -3a, respectively (Table 2). For translation activity, luminescence signals at the peak of transient translation in cells expressing the GND replicon were determined and plotted, together with data from a classical endpoint IRES assay (Figure 4D). The highest translation activity was observed from the GT-1b construct, followed by GT-3a and -2a.

The effects of RNA mutations within the 5′ UTR of the replicon on viral translation and replication were further analyzed by the real-time monitoring system in which those steps can be independently determined in a single culture. Two sites in the HCV IRES within the 5′ UTR are known to be involved in miR-122 binding in hepatocytes [12]. However, evidence on how miR-122 is responsible for HCV translational control as a key regulator for the viral lifecycle is still limited. In the M1 mutant (WT-M1), replication activity was reduced to the detection limit in both HuH-7T1 and Huh7.5.1-8 cells; the M2 mutant (WT-M2) also showed a marked decrease in replication activity, but like the M1 mutant, replication activity was almost completely lost in HuH-7T1 cells, whereas replication activity in Huh7.5.1-8 cells was retained at about 50% of that of WT replicon. Only a little inhibition of the translation peak signals by the mutations was detectable in both cell lines (Figure 5B and Table 3). The limited influence on translation activity by the M1 or M2 mutations was confirmed with SGR/GND (Figure 5C). Using our system, which continuously and individually measures the process from translation to RNA replication of HCV, we confirmed that mutations to the miR-122 binding sites in the 5′ UTR primarily reduce replication activity rather than IRES-dependent translation. Findings support that mutations in the miR122 binding sites have a stronger effect on genome replication than IRES-dependent translation.

## 3. Discussion

Detailed time course analysis of viral lifecycles is important for determining the dynamics of each step in the lifecycles and how their steps are connected and transitioned. To address these issues, conventional virology experiments have analyzed changes in viral expression by collecting cell culture samples over time at intervals of several hours or longer from cells with virus infection or replication. In studies of various RNA viruses, including HCV, viral genome designs with reporters, such as luciferase or fluorescence protein genes, have been widely used for time course analysis of the process from viral translation to genome replication. The reporter assays are highly useful experimental systems that can monitor expression of viral genes/proteins using the reporter activities as indicators. To analyze IRES-dependent translational activity, the bicistronic reporter constructs, which are prepared by inserting the viral IRES element, are commonly used. However, all of these to date have been endpoint assays in which cells or culture supernatants are collected several times a day over time to measure activity [13,14,15,16,17]. In this study, we established an experimental system that can monitor viral replication in cells at 30 min intervals for more than 10 days without cell sampling and found that this real-time monitoring system is useful for analyses with HCV replicons. The subgenomic replicon replication system was capable of monitoring kinetically the transition from initial translation to RNA replication after viral RNA electroporation into cells. A comparison of bioluminescence profiles from the wild-type and RdRp-mutated replicons, in combination with the patterns seen upon the addition of LAP peptide (an HCV translational inhibitor) and DCV (a DAA-targeting HCV NS5A), allowed us to determine which signal peaks reflected the translation and replication processes, respectively (Figure 2A,B).

We found that transient translation proceeds with a peak at 3–4 h post-subgenomic RNA electroporation (hpe), and RNA replication begins to be detected around 20 hpe (Figure 1B). Since RNA replication initiates after a certain interval of time after transient translation reaches its peak, it is likely that not only the production of viral proteins but also the development of an intracellular environment suitable for viral replication are required for replication initiation. The maturation of viral non-structural proteins and their interaction with host-cell factors, presumably at the ER-derived membrane, may reflect the time at which the scaffold for replication and the replication complex are formed.

The time from RNA introduction to the start of RNA replication and the initial speed of replication were shown to be diverse among the HuH-7 subclones, and as expected, the cell lines with higher replication capacity tended to have a shorter time to the start of replication and a higher initial rate of replication. Since the difference in translation efficiency among the four cell lines used was not as much as the difference in replication efficiency, the differences in time to replication initiation and replication rate may not be due to differences in the expression level of the viral proteins, but rather to providing an intracellular environment suitable for viral replication. It is thus likely that the expression levels of host factors that positively or negatively regulate HCV RNA replication differ significantly among the cell lines. It is also possible that the stability of replication regulators may differ among cell lines, as seen in Huh7.5.1-8 cells, where the viral replication transiently increases in the early phase (24–48 h after RNA electroporation; hpe) followed by a rapid decline, and in Huh7T1 cells, where a moderately high level of replication is sustained over a longer period of time. The excessive accumulation of viral gene products in Huh7.5.1-8 cells may result in an intracellular environment unsuitable for viral replication. We are planning to identify regulatory factors that characterize each cell line based on both transcriptome and proteome analyses of cellular factors at not only ~20 hpe but 72–96 hpe, followed by elucidating differences in regulatory mechanisms among cell lines through knockdown and ectopic expression of individual factors. A detailed comparison of the intracellular environment between distinctive cell lines within 24 hpe will help to elucidate the factors that are important for susceptibility to HCV replication. In Huh7.5.1-8 cells, which supported the highest initial replication rate, the signal level at the peak of replication was about 10-fold higher or more than that of other cell lines used in this study, but after reaching the peak, the signal level was maintained for a short period of time and then switched to a declining phase. Although no obvious cell death was observed during replication experiments in Huh7.5.1-8 cells, the excessive accumulation of viral gene products may lead to an intracellular environment unsuitable for virus replication. It is noted that this replication profile with a sharp peak of replication was not due to depletion of luciferase substrate, as the results were the same even when the substrate was added every other day. The kinetically monitored bioluminescence patterns indicating RNA replication of HCV were in good agreement with the temporal changes in NS5A protein expression in HuH-7T1 cells carrying the viral replicon (Figure 1C). In contrast, the change in the number of DMV-containing cells after RNA electroporation (Figure 3A,B) is consistent with the change in replication level at the early phase of replication, but not necessarily at later phases, such as 160–190 hpe. The number of DMV-containing cells decreased significantly, even though the replication level remained high in the replicon cells. The profile of HCV RNA replication in HuH-7T1 cells was bimodal under certain culture conditions (Figure 1B), with the second peak seen after 78 hpe, which was not necessarily accompanied by an increase in the number of cells containing DMVs (Figure 3B).

This is the first study to develop a system for real-time recording of luciferase activity as an indicator of cellular changes over time in a viral replication analysis system without the need to collect replicating cells. The system requires no extra intervention by the experimenter for data collection, minimizes artifacts generated during cell manipulation and sample preparation, and accurately performs continuous multiple measurements. Taking advantage of this feature, this system should be useful not only for replication analysis but also for analysis of the entire life cycle of various viruses, including the early stages of viral infection, and for analysis of temporal environmental changes on the host cell side associated with infection, such as stress responses.

## 4. Materials and Methods

### 4.1. Cell Culture

Human hepatoma HuH-7-JCRB (obtained from the Japanese Collection of Research Bioresource Cell Bank, Osaka, Japan), HuH-7-Scr and Huh7.5.1 (gifted from Dr. Francis Chisari, The Scripps Research Institute, La Jolla, CA, USA), Huh7.5.1-8 [7], and HuH-7T1 [8] cells were maintained in Dulbecco’s modified Eagle medium (DMEM) supplemented with nonessential amino acids, 100 U of penicillin/mL, 100 µg of streptomycin/mL, and 10% fetal bovine serum (FBS) at 37 °C in a 5% CO_2_ incubator.

### 4.2. Reagents

DCV was a gift from Bristol-Myers Squibb (New York, NY, USA). Rabbit polyclonal antibody against NS5A (TB0705) was developed by immunization with the recombinant NS5A protein [18]. Mouse monoclonal antibodies against GAPDH (6C5) was obtained from Santa Cruz Biotechnology (Santa Cruz, CA, USA). Wild-type and mutated La peptides (WT LAP and Y23Q LAP) [5] were synthesized by Eurofins (Tokyo, Japan).

### 4.3. Plasmids

Subgenomic HCV replicon plasmids pSGR-JFH1/WT/SLG (HCV GT-2a), pSGR-Con1/WT/SLG (GT-1b), and pSGR-3a/WT/SLG (GT-3a), which carry the SLG gene as a reporter, were constructed by substitution of an amplified fragment containing SLG from pSLG-HSVtk (Toyobo, Osaka, Japan) into pSGR-JFH1/FLuc [10,19], pSGR-Con1/GLuc [16], and pS310/R2895K [11]. The GDD-to-GND substitution mutation was introduced in the NS5B region to abolish RNA polymerase activity, resulting in pSGR-JFH1/GND/SLG. pRL/1a-Luc (H77 IRES; GT-1a) and pRL/EMCV-Luc (EMCV IRES) were constructed as previously described [20]. The pRL/1b-Luc, pRL/2a-Luc, and pRL/3a-Luc were created by replacing the HCV IRES sequence of pRL/1a-Luc with a synthesized IRES DNA fragment (Integrated DNA Technologies, Coralville, IA) of GT-1b, -2a, and -3a, respectively. In pSGR-JFH1/WT-M1/SLG (WT-M1) and pSGR-JFH1/WT-M2/SLG (WT-M2), the first miR-122 target site (ACACUCC, nt 21–27) and the second target site (CACUCC, nt 37–42), respectively, were mutated to ACUCUCC and CUCUCC (Figure 5A). The primers used for plasmids construction are listed in Appendix A.

### 4.4. RNA Synthesis, Electroporation, and Transfection

RNA synthesis was performed as previously described [16]. Briefly, for linearization, pSGR-JFH1/SLG, pS310/R2895K and pSGR-JFH1/WT-M1/SLG (WT-M1), and pSGR-JFH1/WT-M2/SLG (WT-M2) were digested with *Xba*I, and pSGR-Con1/SLG was digested with *Sca*I. Linearized DNA fragments were then purified and used as templates for RNA synthesis. HCV RNA was synthesized in vitro by a MEGAscript T7 kit (Ambion, Austin, TX, USA). Synthesized RNA was treated with DNase I, followed by acid phenol extraction to remove remaining template DNAs. One microgram of reporter replicon RNA was mixed with 10 µL of cell suspension (1 × 10^5^ cells) in BTXpress buffer (BTX, Holliston, MA, USA) and electroporated by Neon (Thermo Fisher Scientific, Rockford, IL, USA) with electroporation conditions of 1400 V, 20 ms, and 1 pulse. After electroporation, the cells were immediately seeded into 35 mm dishes.

### 4.5. Real-Time Detection of Bioluminescence in Cell Cultures

For real-time measurement of bioluminescence, the culture medium for test cells in 35 mm dishes was replaced with DMEM/10% FBS supplemented with 3 µM D-luciferin (Wako) as a substrate for SLG luciferase. Bioluminescence signals were recorded for 1 min at intervals of 29 min under a 5% CO_2_ atmosphere at 37 °C using a Kronos Dio Luminometer (AB-2550, ATTO, Tokyo, Japan) with the most appropriate filters for each luciferase. Unless otherwise stated, the measurement was continued without exchanging the culture medium.

The sigmoidal functions are fitted to experimental data sets obtained from the real-time measurements. To analyze translation rates, the slopes of the quadratic formulas indicating transient translation were calculated by differentiating each luminescence value with respect to time, with reference to the methods described previously [21]. The slope at the mid-time point between just after the start of measurement and at the time of the highest luminescence value was defined as the initial rate of transient translation (Ri(T)) (Appendix A). The early stage of the viral replication is assumed to be the period from the beginning of the replication process (when the luminescence indicating transient translation reaches a minimum value and then begins to increase) to around 60 hpe. Thus, the slopes of fitted curves during its period were calculated as indicated above, and the slope at the midpoint during the early stage of replication is defined as the initial rate of replication Ri(R) (Appendix A). The sum of luminescence signals during transient translation (Ss(T)) was calculated by summing the signal values detected during the entire period of transient translation (Appendix A). The sum of luminescence signals during replication (Ss(R)) was defined as the sum of signal values detected between the beginning of the replication process and the point at which the value drops to the basal level or 240 hpe (Appendix A).

### 4.6. Immunoblotting

Immunoblotting was performed essentially as previously described [16]. Cell lysates with 1% NP-40, 0.1% SDS, 1% sodium deoxycholate, 25 mM Tris-HCl, pH 7.6, 150 µM NaCl, 1 mM EDTA, and protease inhibitor cocktail (Roche Diagnostics, Basel, Switzerland) were separated by SDS-PAGE and transferred onto polyvinylidene difluoride membranes. After blocking for 2 h, membranes were incubated with the primary antibody against either GAPDH or NS5A for 1 h. After washing, membranes were incubated with horseradish peroxidase-conjugated secondary antibody (Cell Signaling Technology, Danvers, MA, USA) for 1 h. Antigen–antibody complexes were detected using ECL Prime Western Blotting Detection Reagent (GE Healthcare, Little Chalfont, UK).

### 4.7. Cell Growth Measurement

The cell growth curve was generated from measurement of the cell numbers at 24 h intervals for 10 days. At each time point, the numbers of cells were counted in duplicate using a TC20 Automated Cell Counter (Bio-Rad,Hercules, CA, USA), and the mean values were calculated.

### 4.8. RNA Extraction and RT-Quantitative PCR (qPCR)

Procedures for RNA extraction and RT-qPCR were basically described [16]. Briefly, total RNAs were extracted from test cells with TRI reagent (Cosmo Bio, Tokyo, Japan), according to the manufacturer’s instructions. RT-qPCR was performed in the CFX Connect Real-Time System (Bio-Rad) using THUNDERBIRD SYBR qPCR mix (Toyobo, Osaka, Japan). Reverse-transcribed cDNAs together with 6 pmol of forward and reverse primers were used for PCR. The thermal cycling conditions comprised 1 min at 95 °C, followed by 44 cycles at 95 °C for 15 s and 60 °C for 1 min, and fluorescent acquisition at 60 °C, followed by melt curve analysis of temperature increasing from 60 °C to 95 °C with fluorescence readings acquired at 0.5 °C increments. RNA expression data were normalized to that of GAPDH using the comparative threshold method (ΔΔCT).

The OAT1 primers were forward primer 5′-GGAAGCGGGAAGAAGGAGC-3′ and reverse primer 5′-GCTAGTGGCAAACCACAGCA-3′. The GAPDH primers were forward primer 5′-AACAGCCTCAAGATCATCAGC-3′ and reverse primer 5′-GGATGATGTTCTGGAGAGCC-3′.

### 4.9. Flow Cytometry and Cell Viability Assay

Huh7.5.1-8, HuH-7T1, HuH-7-Scr, and HuH-7-JCRB cells were electroporated with in vitro-synthesized RNA using a linearized pBSII-GFP, which carries GFP gene in pBluescript II, in a mixture with BTXpress buffer, followed by seeding 3 × 10^5^ cells into a 35 mm dish. For flow cytometry, the cells were harvested after 48 h culture and resuspended in phosphate buffered saline containing 1% FBS to obtain a single-cell suspension. Flow cytometric analyses were performed using Moflo Astrios (Beckman Coulter, Fullerton, CA, USA). For cell viability assay, the numbers of cells were counted after 2 days of culture in triplicate using a TC20 Automated Cell Counter (Bio-Rad). Cell viability was expressed as the ratio of the number of electroporated cells to the number of cells without RNA electroporation.

### 4.10. Endpoint IRES Assay

Cells were transfected either with pRL/1b-Luc, pRL/2a-Luc, pRL/3a-Luc, or pRL/EMCV-Luc. The cells were lysed at 24 h post-transfection with passive lysis buffer of dual-luciferase reporter assay system (Promega, Madison, WI, USA), and the luciferase activities in cell lysates were measured with a luminescence reader (Synergy H1, BioTek, VT, USA). The dual luciferase reporter assay is characterized by direct correlation of the translation initiation capacity to the *Renilla* luciferase activity. The *Renilla* luciferase light units were normalized by dividing the obtained values by the Firefly luciferase light units.

### 4.11. Ultrastructural Analysis

Cells were fixed with 2.5% glutaraldehyde in 0.1 M phosphate buffer (pH 7.4) for 30 min at room temperature; the cells were then scraped from the plastic plate into a fixative solution in which the cells were incubated for another 90 min at 4 °C. The cells then were post-fixed with 1% osmium tetroxide in 0.1 M phosphate buffer (pH 7.4) for 2 h, embedded in 2% agar, dehydrated in ethanol, embedded in Epon 812, and left to polymerize at 60 °C for 2 days. Ultra-thin sections (50 to 70 nm thick) were picked up on copper grids and were double-stained with uranyl acetate and lead citrate by standard procedures. The specimens were examined by TEM (H-7100; Hitachi Ltd., Tokyo, Japan; JEM1400, JEOL Ltd., Tokyo, Japan). For each sample, one hundred cells were randomly observed and the number of cells in which DMV was detected was counted.

## Figures and Tables

**Figure 1 ijms-23-08711-f001:**
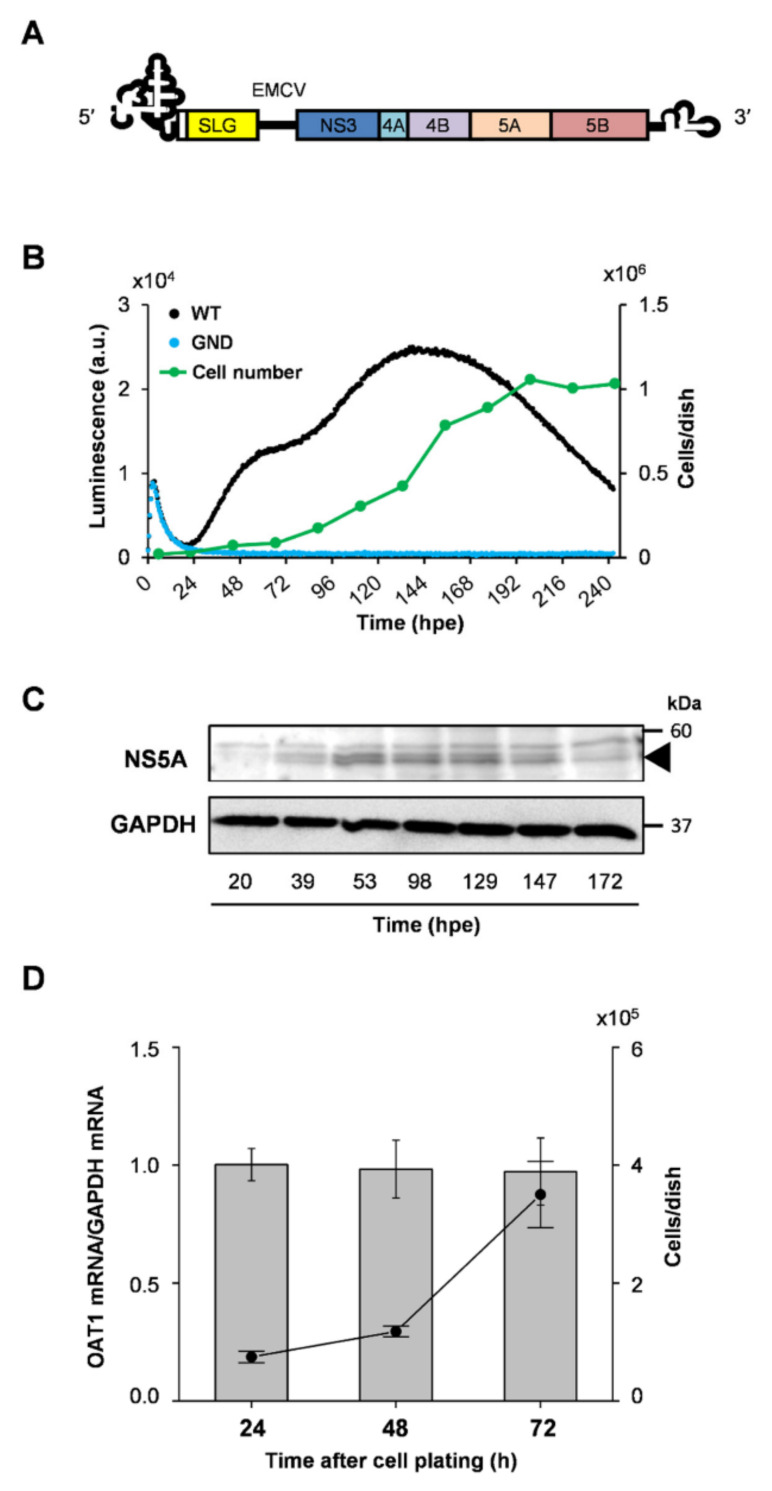
Development of real-time monitoring system of HCV RNA replication. (**A**) Schematic representation of HCV subgenomic replicon used in this study. The subgenomic replicon contains the 5′ untranslated region of HCV, which directs the IRES-mediated translation of the green-emitting luciferase SLG gene. An encephalomyocarditis virus (EMCV)-IRES mediates translation of the HCV nonstructural NS3-NS5B region, which is flanked by the HCV 3′ untranslated region. (**B**) HuH-7T1 cells (1 × 10^5^ cells) were electroporated with SGR-JFH1/WT/SLG RNA (WT; black line) or SGR-JFH1/GND/SLG RNA (GND; blue line). Immediately after electroporation, cells were seeded in a 35 mm dish and luciferase substrate D-luciferin (3 µM) was added to the culture medium. Luciferase activity was chronologically recorded by Kronos Dio. GND represents a mutated replicon defective in HCV RNA replication due to the mutation of GDD motif to GND within NS5B polymerase active site. Cell growth (green line) was measured by counting cell numbers every 24 h after plating of 1 × 10^5^ cells/dish in a 35 mm dish. Results are presented as means (*n* = 2). (**C**) HuH-7T1 cells carrying SGR-JFH1/WT/SLG RNA were harvested at the indicated time points (hpe). Expression of HCV NS5A and GAPDH was analyzed by immunoblotting using anti-NS5A rabbit polyclonal antibody and anti-GAPDH mouse monoclonal antibody, respectively. NS5A p56 and p58 are indicated by an arrowhead on the right. (**D**) HuH-7T1 cells (1 × 10^5^ cells/dish) were seeded in 35 mm dishes. At 24 h, 48 h, and 72 h after cell plating, total RNAs were extracted from cultured cells in each dish. The mRNA expression of OAT1 gene was determined by RT-qPCR. Data normalized to the expression of GAPDH mRNA in the corresponding cultures were represented by bar graphs. Cell growth measured by counting cell numbers was indicated by a line graph. Results are presented as means with SEM (*n* = 3).

**Figure 2 ijms-23-08711-f002:**
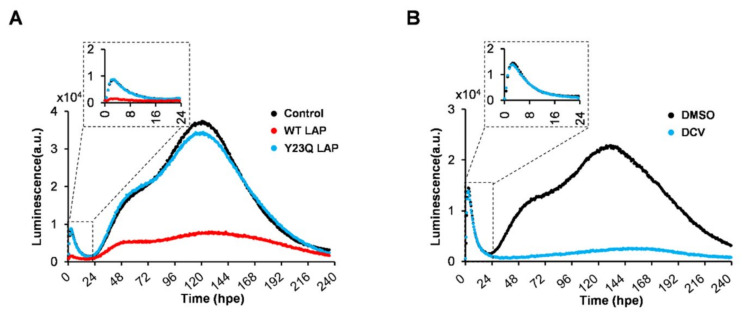
Effects of La peptide (LAP) and NS5A inhibitor daclatasvir (DCV) on the bioluminescence patterns in the replicon system developed. (**A**) SGR-JFH1/WT/SLG RNA was pre-mixed with WT LAP (red line), Y23Q LAP (blue line) or, control (electroporation reagent only; black line), followed by incubating at 37 °C for 1 h and then electroporated into HuH-7T1 cells. Luciferase activity was kinetically recorded by Kronos Dio. Inset: plots of data obtained from the time range of 0 to 24 h are shown. (**B**) SGR-JFH1/WT/SLG RNA was electroporated into HuH-7T1 cells. Immediately after electroporation, the cells were cultured in the medium containing D-luciferin with or without 0.08 nM DCV. Luciferase activity was chronologically recorded by Kronos Dio. Inset: plots of data obtained from the time range of 0 to 24 h are shown.

**Figure 3 ijms-23-08711-f003:**
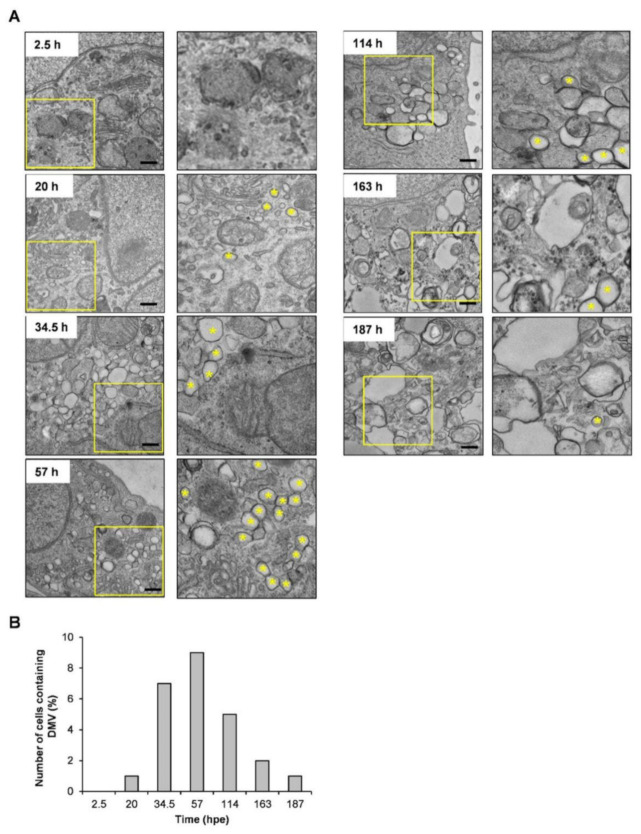
Time course of membrane alterations after electroporation of HCV replicon RNA. (**A**) HuH-7T1 cells were electroporated with SGR-JFH1/WT/SLG RNA, subjecting to TEM analysis at the indicated time points post-electroporation. Double-membrane vesicles (DMVs) in the box with yellow lines are marked with yellow asterisks. Scale bars: 400 nm. (**B**) Number of cells containing DMVs (at least one DMV) was counted in the photos taken by TEM at the indicated time points.

**Figure 4 ijms-23-08711-f004:**
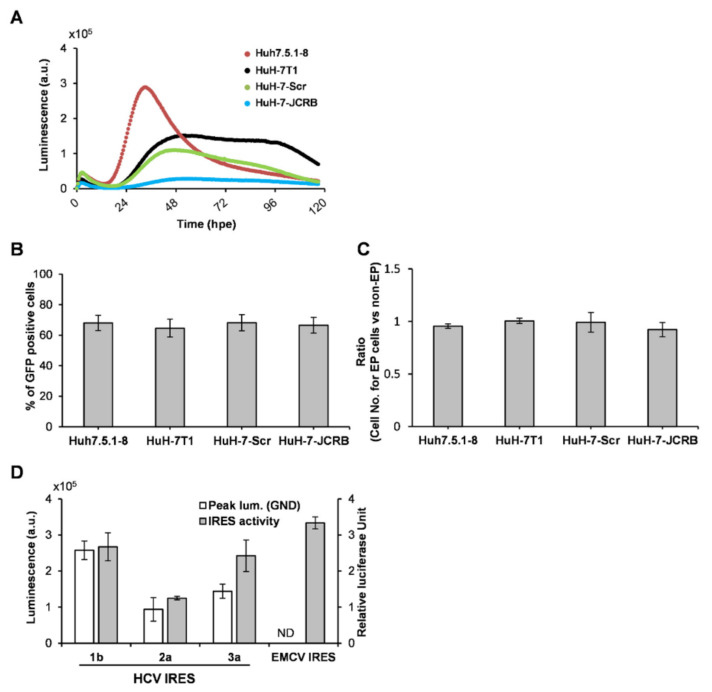
(**A**) HCV replication in several HuH-7-derived cell lines. Huh7.5.1-8, HuH-7T1, HuH-7-Scr, and HuH-7-JCRB cells were electroporated with SGR-JFH1/WT/SLG RNA. After electroporation, 3 × 10^5^ cells were immediately seeded into a 35 mm dish for real time monitoring. Luciferase activity was chronologically recorded by Kronos Dio. (**B**) Efficiency of RNA electroporation into each cell line. As indicated in Appendix A, the percentages of GFP-positive cells in cells electroporated with GFP RNA were determined by flow cytometry. Each bar graph is represented from triplicate measurements (means with SD) for each cell line. (**C**) Effect of RNA electroporation on cell viability in each cell line. The numbers of cells with or without RNA electroporation were counted after 2 days of culture. Cell viability was expressed as the ratio of the number of electroporated cells to the number of cells without RNA electroporation (means with SD, *n* = 3). (**D**) Huh7.5.1-8 cells were transfected with bicistronic reporter pRL/1b-Luc, pRL/2a-Luc, pRL/3a-Luc or pRL/EMCV-Luc. After 24 h, activities of Firefly luciferase (FL) and *Renilla* Luciferase (RL) in cell lysates were determined. The ratio of activity of FL to RL was defined as relative luciferase unit (blank bar) and was compared with the luminescence value at the peak of transient translation of GND replicon obtained in the real time monitoring system (gray bar). Results are presented as means with SD (*n* = 3). ND; not determined.

**Figure 5 ijms-23-08711-f005:**
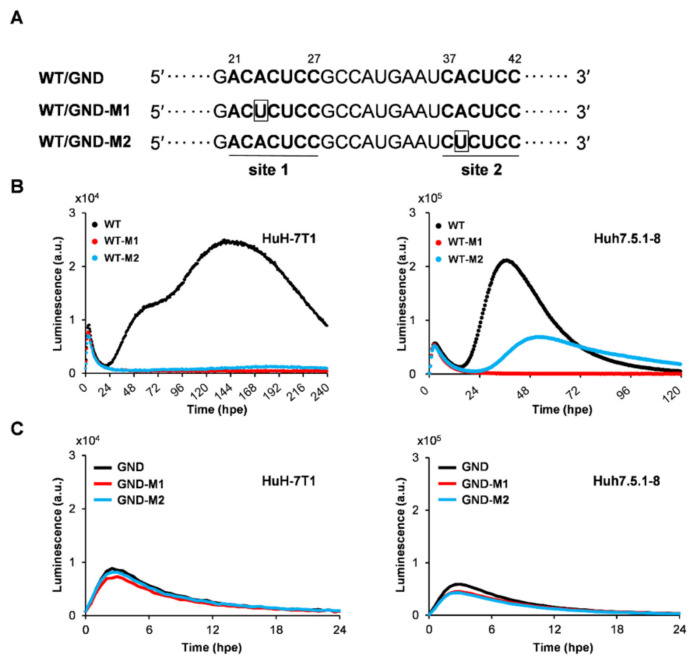
Effects of substitution mutations in the miR-122 binding sites on translation and replication of HCV. (**A**) Schematic representation of the two miR-122-binding sites, site 1 and site 2, located in the 5′ UTR of the HCV genome. The miR-122 complementary sequences in HCV RNA are highlighted in bold. The mutated sequences in WT/GND-M1 and M2 are boxed. (**B**,**C**) HuH-7T1 and Huh7.5.1-8 cells were electroporated with SGR-JFH1/WT/SLG (WT), SGR-JFH1/WT-M1/SLG (WT-M1), and SGR-JFH1/WT-M2/SLG (WT-M2), SGR-JFH1/GND/SLG (GND), SGR-JFH1/GND-M1/SLG (GND-M1), SGR-JFH1/GND-M2/SLG (GND-M2) RNAs. Luciferase activity was chronologically recorded by Kronos Dio.

**Table 1 ijms-23-08711-t001:** Comparison of bioluminescence profiles of transient translation and replication in HuH-7 derived cell lines.

Cell Line	Transient Translation		Replication			
	Ri(T) ^a^	Ss(T) ^b^	Ri(R) ^c^	Ss(R) ^d^	Highest Activity ^e^	
	(a.u./h)	(Area)	(a.u./h)	(Area)	Time (hpe)	Lum (a.u.)
HuH-7T1	10,227	193,944	3556	11,451,313	51.5	151,541
Huh7.5.1-8	18,236	358,071	10,620	10,725,355	33.5	289,386
HuH-7-Scr	18,391	336,048	2728	6,683,591	46.5	110,041
HuH-7-JCRB	6822	106,151	559	1,948,137	52.5	27,892

^a^ Initial rate of transient translation; ^b^ Sum of luminescence signals during transient translation; ^c^ Initial rate of replication; ^d^ Sum of luminescence signals during replication; ^e^ Time point (hpe) in the replication process when the highest luminescence value was obtained and the value (a.u.) at that time.

**Table 2 ijms-23-08711-t002:** Comparison of profiles of transient translation and replication among HCV genotypes.

HCV Genotype	Transient Translation		Replication		
	Ri(T) ^a^	Lum ^b^	Ri(R) ^c^	Lum ^e^	Time ^d^
	(a.u./h)	(a.u.)	(a.u./h)	(a.u.)	(hpe)
1b	122,959	412,961	176	23,399	70.5
2a	45,245	264,026	62,591	748,140	33.5
3a	7005	20,345	3	1028	99.5

^a^ Initial rate of transient translation; ^b^ Luminescence value at the peak of transient translation; ^c^ Initial rate of replication; ^d,e^ Time point (hpe) in the replication process when the highest luminescence value was obtained and the value (a.u.) at that time.

**Table 3 ijms-23-08711-t003:** Effects of mutations within 5′ UTR of the replicon on bioluminescence profiles of transient translation and replication.

Cell Line	Replicon ^a^	Transient Translation			Replication		
		Ri(T) ^b^	Ss(T) ^c^	Lum ^d^	Ri(R) ^e^	Ss(R) ^f^	Lum ^g^
		(a.u./h)	(area)	(a.u.)	(a.u./h)	(area)	(a.u.)
HuH-7T1	WT	3447	77,647	9031	376	3,615,588	12,527
	WT-M1	2974	65,383	7802	<1	105,912	493
	WT-M2	2712	61,398	7090	4	198,529	1346
Huh7.5.1-8	WT	23,439	454,583	57,147	10,798	7,997,501	211,445
	WT-M1	22,291	377,372	54,286	<1	111,366	1138
	WT-M2	21,357	395,020	52,081	2717	3,939,265	69,136

^a^ WT; SGR-JFH1/WT/SLG, WT-M1; SGR-JFH1/WT-M1/SLG, WT-M2; SGR-JFH1/WT-M2/SLG; ^b^ Initial rate of transient translation; ^c^ Sum of luminescence signals during transient translation; ^d^ The highest luminescence value detected during the transient translation process; ^e^ Initial rate of replication; ^f^ Sum of luminescence signals during replication; ^g^ The highest luminescence value detected during the replication process.

## Data Availability

The data presented in this study are available on request from the corresponding author.

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
