# Peer review of "Development and Use of a Kinetical and Real-Time Monitoring System to Analyze the Replication of Hepatitis C Virus"

_ijms, 2022, doi:10.3390/ijms23158711_

Round 1
Reviewer 1 Report
This article shows a viral system that uses a reporter gene to study viral replication. The authors do several claims that this is the first time it is done (see lines 48 and 49), however, I have listed just a few articles were viral replication is studied using different types of reporter systems:1) 10.3390/v8070179
2) 10.1016/S0168-1702(02)00047-3
3) 10.1016/j.jviromet.2005.01.016
4) 10.1016/j.virol.2005.01.021
5) 10.1002/9780471729259.mc15g04s29
6) 10.1016/j.virol.2016.07.015
7) 10.1016/j.jviromet.2005.10.016
8) 10.1002/rmv.1963
9) 10.1089/aid.1994.10.295
10) 10.1016/j.antiviral.2011.05.001
11) 10.3390/v12121457
12) 10.1021/acs.biochem.5b00623
13) 10.1128/AAC.49.12.4980-4988.2005
14) 10.1016/j.rvsc.2014.07.007
Hence, please remove any claim that this is the first time is done.
The idea of circadian rhythms is completely unrelated to the problem in question. I would suggest modifying the first paragraph of the introduction; it has nothing to do with the viral life cycle. Circadian rhythms do not control viral replication.
In several places they use the term replication velocities; however, the proper term would be replication rates. Furthermore, they claim to have calculated this rate, but there are no such calculations; there are only kinetic plots. They need to either remove this statement to calculate the slope of the kinetic graphs (the slope of these graphs is the rate). However, if they do this they will need to give a model of why there are different rates during the experiment.
In lines 84-87 they claim that the kinetic analysis is analogous to endpoint analysis, then what is the advantage of measuring the kinetics?
In lines 126-128 refer to figure 3C but there is no mention of using GFP in the text. Please describe the experiment. Same for figure 3D and lines 134-135.
Please add a discussion of why HCV replication differs depending on the HuH-7-derived cell line. What are the key differences between these cell lines? Is this effect described somewhere else? The analysis of this observation in the discussion section is insufficient. This section is missing RT-qPCR data. Please compare the expression level of the viral RNAs as a function of time in the different cell lines to show that the replication in these cell lines is similar (or not).
In figure 4a the way the utr is written is consistent. Write the complete sequence and box the mutated nucleotides
The description of the results from figure 5 is missing in the discussion section. Please explain the experiments and the results.
The method and material section are extremely efficient. Please list all primers used for all the mutations and better describe how the clones were made and verified. The TEM section needs to be better described.
Author Response
We are grateful to the reviewer for the critical comments and useful suggestions that have helped us to improve our paper considerably.
Point-by-point responses to each comment and the revised manuscript are indicated in the attached file.

Reviewer 2 Report
Suzuki2022 peer
The authors provided a novel cell culture method to analyze HCV replication in vitro in a time-dependent and real-time assay. The overall design of the experiments and writing is good, and I would therefore recommend to address the following minor issues:
In the abstract the authors mentioned that they were the first to develop a real-time system that kinetically detects reporter activity over time without subsampling cells or culture supernatants to study virus replication.
I would like the authors to highlight other methods, their pros and cons in the introduction and/or discussion to help the reader understand the uniqueness of this approach.
In line 37 you mentioned the luciferase half life, could you try to quantify this?
In line 103 you cite unpublished data – it would be great if the authors could add this data to the manuscript, either in the main text or supplement.
The authors cited table 1 to 3 in their manuscript. However, no table was supplemented with the manuscript.
In the sentence starting in line 150 the authors mentioned the benefits of their in vitro system and how miR-122 effects RNA replication. The following sentence does in my opinion not add much additional information to the manuscript. I would suggest connecting the statement with the IRES dependent translation with the preceding sentence
Line 162: and throughout the manuscript: 1x105 cell should be 1x105
Line 168: n = 3 would be preferred, SD/SEM should not be calculated from 2 data points
Line 201: Site1 should be site 1 (include space)
In general, the resolution of many figures could be improved.
Standard deviation (SD) is preferred over standard error of the mean (SEM).
Figure 1B
Please provide a third replicate. And add the standard error to the graph as confidence interval, which allows the reader to access reproducibility.
Figure 1C
Please provide information on how often this experiment was conducted and whether it was reproducible.
Figure2
Please indicate the number of replicates and their SD/SEM.
Figure 3A
Please show the number of replicates and SD/SEM
Figure 3BCD
In subfigure B/C error bars are indicated as +/- SEM, while in subfiure D only + SEM is indicated. Even though the white bars were normalized as RLU, they could show SD/SEM
Figure4
Please indicates n and SD/SEM statistics.
Figure5
Not references in the main text.
B: How many images were taken? Were they specifically selected for or were the positions determined randomly?
Discussion:
Please provide more details on other cell culture methods allowing to track HCV replication and translation and how your method relates to that. What does your method set appart from previous literature?
Author Response

(The authors gave the same response as above.)

Round 2
Reviewer 1 Report
I appreciate the work done by the authors to address all my observations. The manuscript has been greatly improved, and it is almost ready to be published. However, there is only one thing missing that is very important; the equations to calculate the rates that are in tables 1 - 3. Please add the rates and the basic assumptions needed to make the kinetic model. This would be the only modification I have to ask.
Author Response
We are grateful to the reviewer for the important suggestion. Our response to the comment is indicated in the up-loaded file with the second revised manuscript.
